# Inflammatory Bowel Disease: Clinical Diagnosis and Surgical Treatment-Overview

**DOI:** 10.3390/medicina58050567

**Published:** 2022-04-21

**Authors:** Amosy Ephreim M’Koma

**Affiliations:** 1Department of Biochemistry, Cancer Biology, Neuroscience and Pharmacology, Meharry Medical College School of Medicine, Nashville, TN 37208-3500, USA; amkoma@mmc.edu or amosy.e.mkoma@vumc.org; Tel.: +1-615-327-6796; Fax: +1-615-327-6440; 2Department of Pathology, Anatomy and Cell Biology, Meharry Medical College School of Medicine, Nashville General Hospital, Nashville, TN 37208-3599, USA; 3Division of General Surgery, Section of Colon and Rectal Surgery, Vanderbilt University School of Medicine, Nashville, TN 37232-0260, USA; 4The American Society of Colon and Rectal Surgeons (ASCRS), 2549 Waukegan Road, #210, Bannockburn, IL 600015, USA; 5The American Gastroenterological Association (AGA), Bethesda, MD 20814, USA

**Keywords:** inflammatory bowel disease, ulcerative colitis, Crohn’s colitis, indeterminate colitis, colitis-associated colorectal cancer, clinical diagnosis guideline, diagnostic challenges, molecular diagnostics advances, surgical treatment guidelines, uneven representation of socioeconomic strata, IBD care during COVID-19 pandemic

## Abstract

This article is an overview of guidelines for the clinical diagnosis and surgical treatment of predominantly colonic inflammatory bowel diseases (IBD). This overview describes the systematically and comprehensively multidisciplinary recommendations based on the updated principles of evidence-based literature to promote the adoption of best surgical practices and research as well as patient and specialized healthcare provider education. Colonic IBD represents idiopathic, chronic, inflammatory disorders encompassing Crohn’s colitis (CC) and ulcerative colitis (UC), the two unsolved medical subtypes of this condition, which present similarity in their clinical and histopathological characteristics. The standard state-of-the-art classification diagnostic steps are disease evaluation and assessment according to the Montreal classification to enable explicit communication with professionals. The signs and symptoms on first presentation are mainly connected with the anatomical localization and severity of the disease and less with the resulting diagnosis “CC” or “UC”. This can clinically and histologically be non-definitive to interpret to establish criteria and is classified as indeterminate colitis (IC). Conservative surgical intervention varies depending on the disease phenotype and accessible avenues. The World Gastroenterology Organizations has, for this reason, recommended guidelines for clinical diagnosis and management. Surgical intervention is indicated when conservative treatment is ineffective (refractory), during intractable gastrointestinal hemorrhage, in obstructive gastrointestinal luminal stenosis (due to fibrotic scar tissue), or in the case of abscesses, peritonitis, or complicated fistula formation. The risk of colitis-associated colorectal cancer is realizable in IBD patients before and after restorative proctocolectomy with ileal pouch-anal anastomosis. Therefore, endoscopic surveillance strategies, aimed at the early detection of dysplasia, are recommended. During the COVID-19 pandemic, IBD patients continued to be admitted for IBD-related surgical interventions. Virtual and phone call follow-ups reinforcing the continuity of care are recommended. There is a need for special guidelines that explore solutions to the groundwork gap in terms of access limitations to IBD care in developing countries, and the irregular representation of socioeconomic stratification needs a strategic plan for how to address this serious emerging challenge in the global pandemic.

## 1. Background

Colonic inflammatory bowel disease (IBD), or the colitides, encompasses Crohn’s colitis (CC) and ulcerative colitis (UC), the two highly heterogeneous, debilitating, incurable, persistent, relapsing/worsening, immune-arbitrated inflammatory pathologies of the digestive system canal [1]. UC causes inflammation and ulceration of the epithelial layer and, to a lesser degree, the submucosae layer of the larger intestine (colon and rectum) only [2]. CC varies from UC in location, in that it is segmental and causes inflammation impacting the whole digestive system from the mouth to the anus and can further cause inflammation deeper within all the intestinal layers (transmural and skip lesions) that may affect other organs through fistulation [3,4]. IBD has significant impacts on patient health quality of life (QoL), mental health, work productivity, and healthcare resources [5,6].

There are established guidelines for the diagnosis of IBD [7,8], which include international clinical practice tool recommendations that incorporate various best practices, and other evidence has widely been issued [9,10]. Thirty percent of IBD patients with colonic IBD present with ambiguous diagnosis [11]. In the past two decades, there have been vast advances in research, i.e., molecular diagnostics and surgical technical evolution for IBD management [12,13]. The aim of this overview is to provide disease guidance consensus for healthcare professionals managing IBD, to ensure that investigation, diagnosis, surgical treatment, and monitoring decisions are based on the best available common consent evidence, and to promote and ameliorate the best accepted practice.

Effectuates of IBD are not yet fully understood, but are believed to be multifactorial [1,14,15], i.e., a susceptible host (e.g., genetic, gut barrier and the exaggerated innate/adaptive immune response) and external/exogenous factors (e.g., normal indigenous intestinal luminal flora) are important basic associates that probably induce and perpetuate the pathogenesis of IBD [16]. The mechanistic trigger processes are mediated through components of the autoimmune response to self-antigens [17,18]. Recently, research has paid attention to the role of antibodies in downstream events and mechanisms of autoimmunity and inflammation [19,20]. Whether the fabrication of antibodies is a serologic product of IBD or if it is a consequence of barrier dysfunction caused by inflammation remains a significant knowledge gap.

While the Western world including US, Canada and Europe continues to advance and improve ambulatory regimens care delivery [21,22] to meet high-quality, safety, efficacy, coordination of care, and recommended precision evidenced-based care in IBD patients [23,24], developing nations at large have healthcare service constraints and limitations to meet the required standard of care [25] due to limited resources and healthcare personnel not being trained and having no knowledge about treating these diseases [26]. Furthermore, regarding cost-effectiveness considerations and recommendations by the World Health Organization (WHO) and the World Gastroenterology Organization (WGO) [27,28], developing countries struggle the most. The economic implications of IBD are enormous [29]. Hospital admission rates and costs for IBD show an increasing trend [30,31]. In the US alone, the estimated annual direct treatment costs are greater than USD 6.8 billion, and indirect costs amount to an additional USD 5.5 billion [32,33]. The healthcare systems, both at the level of primary care and referral hospitals in developing nations, face significant infrastructural limitations as they lack the regular clinical supervision and laboratory assessments needed for evaluating, diagnosing, treating, and monitoring IBD patients [5,6], and will increasingly have difficulty affording the surgical treatment need of these patients as per the herewith presented approval guidelines compared to developed wealthy societies.

## 2. Methods

A literature search for the diagnosis and treatment recommendation guidelines for IBD was performed using predetermined protocols from PubMed, Cumulative Index of Nursing and Allied Health Literature (CINAHL), the Google search engine, Cochrane Database and IBD-associated society organizations, i.e., the American Gastroenterological Association (AGA), the American Society for Gastrointestinal Endoscopy (ASGE), the British Society of Gastroenterology (BSG), the International Foundation for Gastrointestinal Disorders (IFGD), the American Society of Colon and Rectal Surgeons (ASCRS), the American College of Gastroenterology (ACG), the Society of American Gastrointestinal and Endoscopic Surgeons (SAGE), the International Organization for the Study of Inflammatory Bowel Disease (IOIBD), the World Health Organization (WHO), the United States Food and Drug Administration (USFDA), the European Medicines Agency (EMA), European Crohn’s and Colitis (ECC), American Crohn’s and Colitis (CCFA), the Canadian Association of Gastroenterology (CAG), and in accordance with the quality of reporting, meta-analyses of observational studies (MOOSE) [34,35], MEDLINE and EMBASE were searched between 1980 and 2021.

## 3. Clinical Diagnosis and Manifestation

Currently, there is no standardized diagnostic test tool for IBD [36,37]. The standard state-of-the-art diagnosis of IBD relies on amassing of clinical, radiologic, endoscopic, and histopathologic clarification [38,39]. This inexact compilation technique is not always accurate, and about 15% of colonic IBD patients cannot be delineated as UC or CC and are labeled as having ‘‘indeterminate colitis’’ (IC). This is because the clarification criteria for UC and CC are indefinite [40,41]. In addition, another 15% of the colonic IBD cases that undergo pouch surgery, i.e., restorative proctocolectomy with ileal pouch-anal anastomosis (RPC-IPAA) for their definitive UC diagnosis based on the pathologist’s final designation of endoscopic biopsies, will have their initial UC diagnosis reciprocated to ileal Crohn’s disease (CD) based on the postoperative follow-up when clinical and histopathology changes indicate the evolution of CD in the ileal reservoir and/or because authentic CC was not evident prior to colectomy [42,43]. Half of these patients with pouch ileal CD will require reservoir/pouch excision or diversion [44,45].

### 3.1. Ulcerative Colitis

Ulcerative colitis’ (UC) peak onset is mostly in early adulthood [46]. A consequence of untreated UC is chronic inflammation and ulcerations in the mucosal and to a lesser degree submucosal linings confined to the large intestine (colon and rectum) [39,46]. Approximately 15% of patients may encounter hostile development, and these patients may require hospital admission for fulminant disease [46,47]. To establish the diagnosis and disease state of a patient sample, gastrointestinal pathologists depend most on nanoscopic visual examination and the elucidation of marked and/or colored tissue sections [48,49]. These procedures are endowed with a significant degree of discourse [50], and are surfeited with expostulations [50,51]. Careful professional tutoring in pathology subspecialties has helped to achieve the benchmark of care and abolish exorbitant oversights [52,53]. Notwithstanding these eminently thorough benchmarks, ineludible scenes arise in which impartiality cannot be formally assured and where significant variance of opinion occurs amongst consultant specialists [54]. Further to the fundamental guidelines and associated specialized reviews, moderate to severe UC is circumscribed based on the Truelove and Witts criteria and Mayo Clinic score, as presented in Table 1 [55,56,57]. Mayo Clinic scores of 6–12 with an endoscopic subscore of 2 or 3 are viewed as moderate to severe disease. These guidelines are explicated as hospital-admitted patients with the following Truelove and Witts criteria: six or more hematochezia (bloody diarrhea) movements/day with at least one marker of inseparable toxicity, including heartbeat/rate > 90 beats/min, body temperature > 37.8 °C, blood hemoglobin < 10.5 g/dL, and/or an erythrocyte sedimentation rate (ESR) of −30 mm/h [56].

### 3.2. Crohn’s Disease

Predominantly colonic Crohn’s disease, or Crohn’s colitis, is an IBD diagnosed in at least four patients per 100,000 live births in the United States and Canada, and the incidence and prevalence are rising internationally [58,59,60], specifically in developing nations [26,58]. Clinically, CC differs from UC in that it may result in inflammation deeper within the entire colonic walls (mucosa, submucosa, muscularis and serosa, (trammeller) (colon, and rectum) [39]. Furthermore, CC may also affect other systemic organs outside the colon tract through fistulation [3,4,61]. The conciliate features for diagnosing CC comprise an inexact combination of classification systems discussed above in Section 3 for IBD clinical diagnosis, and histopathological findings demonstrating focal, asymmetric, transmural, or granulomatous features [62,63]. Abdominal computed tomography (CT) colonography is the most widely recommended and preferred first-line radiologic study used in the evaluation/assessment of CC. The diagnostic accuracy of magnetic resonance colonography is equivalent to that of CT scans and prevents liability exposure to ionizing radiation. Endoscopic score metrics are the gold standard tool used to estimate the activity of CC, and they are used more often in clinical trials to compute proof of the efficacy and safety of various drugs inducing and maintaining remission and mucosal healing. There are several multipronged scoring systems, but the most used to measure clinical disease severity include the CC Activity Index (CDAI), the Harvey–Bradshaw index (HBI), the short IBD questionnaire (SIBDQ) and the Lehmann score [62,63].

### 3.3. Indeterminate Colitis

In colonic IBD, delineation between CC from UC is often inconclusive [11,40,41,64], thereby confounding effective and appropriate surgeries [39]. Approximately 30% of patients with colonic IBD are indistinguishable, especially during the prodromal stage, and are therefore labeled as ‘‘indeterminate colitis’’ (IC) due to the non-definitive establishment of criteria for CC and UC [40,65,66]. Therefore, understanding the biomolecules and different cellular mechanisms driving IBD heterogeneity is vital to the development of future drug inhibitors to improve patient care [67,68,69,70]. The distinction between UC and CC in otherwise IC is of utmost importance when determining a patient’s candidacy for RPC-IPAA, the standard curative surgical procedure in the treatment for UC. The success of RPC-IPAA surgery and convalescence largely depend on correct diagnosis. To address the IBD diagnosis dilemma in clinical settings, there are published data that have shown robust evidence supporting the presence of human alpha defensin 5 (*DEFA5*, alias HD5) in the colon crypt mucosa with aberrant expression of Paneth cell-like cells (PCLCs) and apparent crypt-cell-like cells (CCLCs) in areas identified with an ectopic colonic ileal metaplasia that is consistent with the diagnosis of CC [11,71]. This conceptual innovation relies on the expression of *DEFA5* and the CCLCs in the colonic mucosal crypt of CC patients and its definitive discriminatory use as a biomarker to facilitate the unambiguous diagnosis of CC with a positive predictive value (PPV) of 96 percent [11,71].

## 4. Core Tip

Histopathology and clinical evaluation show that CC and UC, the two major classifications of IBD subtypes, are indeed discrete entities and have disparate causes and distinguishable mechanisms of tissue damage [72,73]. The foundational inquiry is why the systemic innate immune process responds aggressively to indigenous innocuous inextinguishable bacteria (the commensals), deliverance complex mixes of tissue by-product signatures (cytokines, chemokine, growth factors) and other substances that cause inflammation (antibody–antigen reaction against mucosal resistance).

## 5. Surgical Treatment

When diet, health coaching and lifestyle changes, conservative drug therapy, or other treatments fail and do not relieve IBD symptoms, surgery is inevitably recommended.

**Surgery for ulcerative colitis:** One of the greatest achievements in surgery of the colon and rectum over the past four decades has been the development and reconstruction of the sphincter-preserving pelvic pouch operation, the RPC-IPAA (Figure 1), in patients with refractory UC [74,75]. The development and refinement of pelvic pouch surgery requires the excision and removal of the entire diseased colon while maintaining gut continuity, trans-anal fecal continence/defecation, deferral, discrimination, and fertility. The success of RPC-IPAA surgery is largely dependent on careful patient selection, in particular the correct diagnosis combined with meticulous surgical technique [12,76,77]. Pelvic pouch reconstruction surgery, RPC-IPAA, is the criterion standard surgical procedure for patients with UC [74,75]. The 5- and 10-year cumulative risk of colectomy (emergence or elective) is 10–15%, primarily limited to patients with moderate to severe disease activity; a subset of hospitalized patients with acute severe UC (ASUC), also called fulminant UC, have short-term colectomy rates of 25–30% [47,78,79,80,81]. In some cases, it is not possible to maintain a pouch due to subsequent complications; in this case, a permanent terminal ileostomy may be required [44,82,83,84,85,86,87,88].

**Surgery for Crohn’s colitis:** In CD, there is inflammatory mechanical stress process that plays a critical role in intestinal fibrosis and smooth muscle hyperplasia, causing luminal continuity to be impaired and/or blocked [91,92]. Due to bowel strictures and/or obstructions, containing various degrees of inflammation, fibrosis, and hyperplasia formation, up to two-thirds of people with CC will require at least one surgery in their lifetime. However, surgery does not cure CC. Currently, fibrotic strictures require endoscopic balloon dilatation and/or surgical resection. Surgical resection seeks to remove downstream blockages or constriction and keeps the upstream distended bowel to maintain the Lumina continuity, but recurrences after surgical resection occur at a rate of 100% and patients still suffer long-term bowel symptoms, i.e., abdominal discomfort, pain, and constipation. The benefits of surgery for CC are usually temporary, and the disease often recurs, frequently at the anastomosis site. Postoperatively, it is recommended to prescribe adjuvant medication to minimize and/or delay the risk of recurrence, which occurs at a rate of 100%, given sufficient time after resection, and many patients will still suffer long-term bowel symptoms, i.e., constipation and abdominal pain.

**Surgery for indeterminate colitis:** The inability to distinguish between CC and UC leads to the diagnosis of IC [11]. Here, dispersal favors UC, but focal transmural inflammation or inflammation in the ileum is anticipated in blackwash ileitis and no fistulation [11]. IC is seen in about 15% of patients suffering from IBD when features attributable to both UC and CD are inconclusive [11,71]. Dysplasia on a background of IC warrants surgical resection based on the same principles of management for either CD- or UC-associated dysplasia. The specific surgical approach is based on whether UC or CC is more likely. In such a circumstance, human alpha defensin 5 (*DEFA5*) testing is found to be an accurate candidate biomarker to assist in delineating between CC and UC in the IC patient cohort [11,71]. When the disease is more severe, completing the total abdominal colectomy (TAC) first to assist in making a pathological diagnosis may be effective. This can be followed by a proctectomy (in the case of CC) or IPAA (in the case of UC). In cases where UC is suggested, and in a highly select group of patients with a CC-like phenotype without ileal or anal disease, an RPC-IPAA may be considered. Overall, treatment is often based on the likely phenotype and patients should be advised that if ileal-pouch CD manifests itself after an RPC-IPAA, conversion to an end ileostomy is likely [42,44].

**Surgical management of patients with IBD during the COVID-19 pandemic:** The coronavirus disease 2019 (COVID-19) pandemic has been a global tragedy that changed the traditional pharmacologic and surgical management plan of patients with IBD [10,93,94,95]. The main clinical outcomes were maintained during the COVID-19 pandemic period largely because scheduled visits were replaced by phone calls and virtual consultations [96,97,98]. Virtual clinic follow-ups used the contact center service (CCS) based on the reorganization of high-volume IBD centers and on the continuity of care during the COVID-19 pandemic. This approach could be implemented after the pandemic to optimize the resources of IBD centers [96,99]. This has led to substantial changes, causing the interruption of non-essential endoscopic procedures and outpatient visits, particularly impacting the assessment of disease activity by increasing the risk of relapse, disease complications, delays of new diagnosis and the detection of early post-operative recurrence of CD [100]. The interventional performance of routine endoscopy was largely suspended in many IBD clinics and centers worldwide where severe acute respiratory syndrome coronavirus 2 (SARS-CoV-2) had spread [93]. Experts highlight different scenarios in which endoscopy should still be performed imperatively in unique circumstances in patients with IBD, as well as suggested instructions regarding the use of personal protective equipment [93,101] for carrying out safe procedures and the possible risks of postponing endoscopy in IBD and a post-pandemic plan for access to endoscopy as summarized by Iacucci et al. [93]. The Clinical Practice Update (CPU) from experts presented evidence that provides timely council on the surgical treatment of patients with IBD during the COVID-19 pandemic. Admittedly, the comments herewith provide perspective on a topic of high surgical importance that underwent internal peer review by the Clinical Practice Updates Committees and external peer review through standard procedures of gastroenterology [10,93], which are highlighted in Figure 2. We are reminded, however, that as the understanding of the novel coronavirus progresses, IBD-specific issues and guidance may change beyond reasonable doubt [10,102]. 

## Figures and Tables

**Figure 1 medicina-58-00567-f001:**
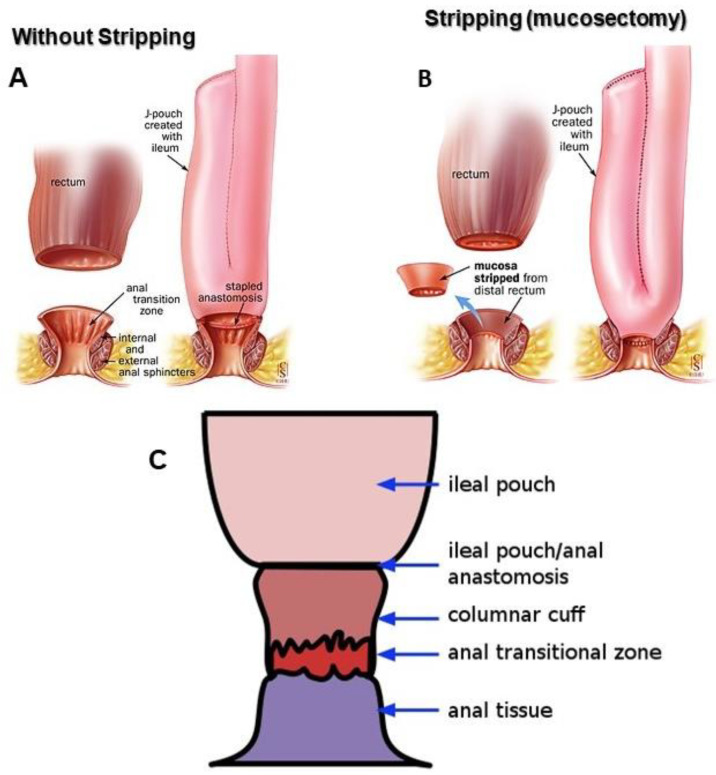
Restorative proctocolectomy with ileal pouch-anal anastomosis. (**A**) J-stapled double staples anastomosis and (**B**) handsewn anastomosis with mucosectomy. Reprinted/adapted with permission from M’Koma et al. [12]. Copyright © 2007, Springer-Verlag. (**C**) Columnar cuff: cuff inflammation is a common complication of RPC-IPAA, especially when a towed anastomosis has been used without mucosectomy. Reprinted/adapted with permission from Ref. [89], Copyrights © 2003 by the American Gastroenterological Association and © 2010 byThieme Medical Publishers, Inc. from Ref. [90].

**Figure 2 medicina-58-00567-f002:**
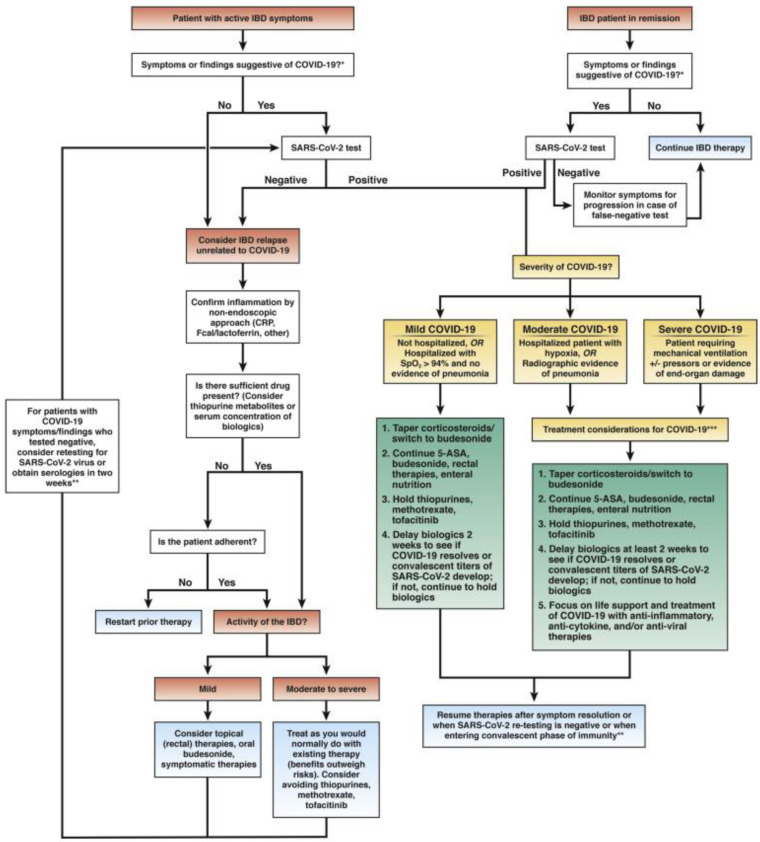
Management of patients with IBD during the COVID-19 pandemic. 5-ASA, 5-aminosalicylic acid medication; CRP, C-reactive protein; mAb, monoclonal antibodies. * Symptoms and consequences of COVID-19: fever (83–99%); cough (59–82%); fatigue (44–70%); anorexia (40–84%); shortness of breath (31–40%); sputum production (28–33%); myalgias (11–35%); headache, confusion, rhinorrhea, sore throat, hemoptysis, vomiting, and diarrhea (<10%); lymphopenia (83%); computed tomography chest: bilateral, peripheral, ground-glass opacities [103]. ** Clearance of SARS-CoV-2 may enable the resumption of IBD therapy; the role of serologic antibody testing is unclear at the current time. *** Indicating new a sentence. Viral clearance testing may or may not be possible or appropriate, given local testing capabilities and health system-approved epidemiological testing strategies during the COIVD-19 pandemic. Treatments for COVID-19 are under investigation, considering therapies that have safety and efficacy in IBD. Reprint /Doneadapted with permission from Rubin et al. Ref. [10].

**Table 1 medicina-58-00567-t001:** Disease severity scoring systems. Reprint/adapted with permission from Refs. [55,56,57]. Copyright © 2020 Elsevier Inc.

	**Truelove and Witts Criteria**				
**Variable**	**Mild**	**Severe**		**Fulminant**	
No. of stools/day	<4	>6		>10	
Blood in stool	Intermittent	Frequent		Continuous	
Temperature, °C	Normal	>37.5		>37.5	
Pulse rates/min	Normal	>90		>90	
Hemoglobin	Normal	<75% normal		Transfusion required	
Erythrocyte sedimentation rate, mm/h	≤30	>30		>30	
Colonic features on radiograph/imaging	None	Air, edematous wall, thumbprinting		Colonic dilatation	
Clinical signs	None	Abdominal tenderness		Abdominal distension and tenderness	
	**Mayo Score for Ulcerative Colitis**				
**Variable**	**Definition**	**Score**	**Variable**	**Definition**	**Score**
Stool pattern	Normal no. of daily bowel movement	0	Endoscopic finding	Inactive colitis	0
	1–2 more bowel movement than normal	1		Erythema, vascularity	1
	3–4 more bowel movement than normal	2		Friability, marked erythema, erosions	2
	5 or more bowel movement than normal	3		Ulceration, severe friability spontaneous bleeding	3
Most severe rectal	None	0	Physician Global	Normal	0
bleeding of the day	Bollo streaks seen in the stool less than	1	Assessment	Mild colitis	
	half of the time				1
	Blood in most stool	2		Moderate colitis	2
	Pure blood passed	3		Severe colitis	

## Data Availability

All data analyzed during this study are included in this published article.

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
