# Peer review of "Inflammatory Bowel Disease: Clinical Diagnosis and Surgical Treatment-Overview"

_medicina, 2022, doi:10.3390/medicina58050567_

Round 1
Reviewer 1 Report
Major revisions:
The review is too long and confusing. Maybe, the author can do separately, two reviews:
- Clinical diagnosis and medical management
- Surgery
In the revision, the authors need to change the following:
The abstract is too long and confusing. The author needs to rewrite it completely.
Please remove the following sentence or explain why you inserted it in the abstract: To date, there is no Food and Drug Administration (FDA) proven bioassay test by which the diseases can be distinguished with certainty.
Why did the authors mention that “This diagnosis inaccuracy poses unique challenges confounding the treatment regimen which is scarcely addressed in extant guidelines”?
Please explain the sentence: “The available diagnostic avenues for either form of IBD to interpret disease severity differs and is determined by geographic divergence in other constrains that burlesques IBD”
We do not use probiotics for preventing surgical complications, so this sentence needs to be removed from the text: “Probiotics have a demonstrable effect in preventing subsequent surgical complications”
The authors need to be more objective: they are talking about clinical diagnosis and treatment overview. They mixed too many things together.
Background: the authors should summarize:
First: introduce IBD, both UC and CD, talking about the clinical impact that both diseases have, including quality of life, impairment to work, etc. (PMID: 31636478, 35239962).
Then you introduce how the diagnosis can be made
Then you talk about the treatment overview. Exclude anything about surgery. This should be focused in another review.
Maybe, it would be interesting for the authors to base themselves on the article PMID 32242028. This is an excellent review about Crohn’s disease.
I recommend focusing only on adults, please exclude the pediatric IBD, it should be focused on another review.
I do think that the immunology is also not the focus of this review, so I recommend exclude it either.
The cost is not the focus of this review either, so please exclude. Maybe, only say about this in the introduction, as a background, but do not include all about costs in the review.
The authors mentioned in the main text very shortly about antibiotics, other medications and supplements: Anti-diarrheal medications: Analgesics: and Vitamins and supplements: Please exclude all. This is not the focus of the clinical diagnosis or treatment
Maybe they can talk about nutrition and anemia. It would be interesting (maybe using one of the following references PMID: 31560755, 33691543 and 33027478 – for anemia; PMID: 33924119
There are many old references, maybe it would be interesting in excluding them.
Finally, add the following reference to talk about COVID and IBD: PMID 33776370 and 33214164
Author Response
I have attached rebuttlal responses

Reviewer 2 Report
In the manuscript submitted by M‘Koma, the author summarized some general information regarding the inflammatory bowel disease. Although some of the content were of interest to researchers/physicians in this specific area, there are some major issues should be addressed:
- It is not clear whom the targeted audiences are. Is this manuscript prepared for the general audience or specialists? The title stated the main content of the text should be clinical diagnosis and treatment, but the information in the related text were way too general.
- Just as mentioned above, the manuscript lacks focus. It is better the author focuses on one or two specific topics and dig deeper. For example, there should be plenty to talk about regarding the molecular advances in diagnosis of IBD.
- The author’s understanding of RMP is kind of misleading. Further, reference 159 and 160 are not the right ones.
- Some typo and uncomprehensive sentences appeared in the text. The manuscript should be carefully edited.
Author Response
I couldn't save the changes and I may have to redo it. But what I sent ie revision. Let me know.
Round 2
Reviewer 1 Report
1. The author did a significant improvement in the manuscript quality. However, there are some issues that need to be fixed. First, why did the author change the manuscript title for "colonic" instead of IBD? It would be more interesting for the journal readers to do a review on clinical diagnosis (of Crohn's and Ulcerative colitis). I had previously suggested to the editor-in-chief to do two manuscripts (One: clinical diagnosis and medical management; Second: overview of surgical treatment). Depending on the editor-in-chief you can keep all the revisions (clinical diagnosis, medical management and surgical treatment) in one single article, however, I do think that it will be too long. I would recommend to separate in two, as mentioned above. 2. Please exclude the term "colonic" and discuss IBD in general, not only colonic IBD. It will be better for the journal readers. 3. Please include the following reference in the manuscript: PMID: 33121688 4. Please use some reference guide, for example Vancouver style, or according to Medicina guidelines. The references do not follow a standard style during the text (pages 10-16). 5..
Author Response
RESPONSE REVISION 2 - medicina-1639493
I truly thank the reviewers for their time committed to go through this paper again and for the new comments. I have responded to the reviewer’s concerns point by point as follows:
Reviewer 1.
The author did a significant improvement in the manuscript quality. However, there are some issues that need to be fixed.
- First, why did the author change the manuscript title for "colonic" instead of IBD? It would be more interesting for the journal readers to do a review on clinical diagnosis (of Crohn's and Ulcerative colitis). I had previously suggested to the editor-in-chief to do two manuscripts (One: clinical diagnosis and medical management; Second: overview of surgical treatment). Depending on the editor-in-chief you can keep all the revisions (clinical diagnosis, medical management, and surgical treatment) in one single article, however, I do think that it will be too long. I would recommend separating in two, as mentioned above.
Response: The reviewer’s suggestion to the editor-in-chief to do two manuscripts (One: clinical diagnosis and medical management; Second: overview of surgical treatment). This suggestion has been implemented and the original long manuscript was separated into two. This article is an overview of surgical treatment.
- Please exclude the term "colonic" and discuss IBD in general, colonic IBD. It will be better for the journal readers.
Response: The term "colonic” has been removed from the title. Please note – Colorectal surgeons are specialized and focused on the Colon and Rectum. We wanted to make it clear about that from the title but as per the reviewer suggestion I have deleted the word.
- Please include the following reference in the manuscript: PMID: 33121688
Response: As recommended by the reviewer, the paper PMID: 33121688 is included are reference No. 57
- Please use some reference guide, for example Vancouver style, or according to Medicina guidelines. The references do not follow a standard style during the text (pages 10-16). 5.
Response: According to the MDPI guidelines, authors are instructed to use the “American Chemical Society (ACS)” style for the references. I used the ACS in this paper as per the MDPI author-instructions.
Reviewer 2.
In the revised version the author addressed some of the concerns raised by this reviewer, and the current version indeed is more focused. Nonetheless, couple suggestions for the author:
- Restructure the text. There are too many headings, and some of them could be combined into one section as subheadings. For instance, after Clinical diagnosis there are CD, UC, IC, etc. Could they be put under “clinical diagnosis and manifestation” or something similar. Also, “treatment and management” could cover several of the current headings.
Response: I thank the reviewer for his suggestion. The IBD subtypes, i.e., UC, CC and IC are under section 3 “clinical diagnosis and manifestation”. All are under Section 3 compilation to make it easier for readers. I did the same for Surgical treatment under section 6.
- In the molecular diagnostic advances section, is there any other new progress? As with the development of multi-omics I am curious if there is any related research in IBD--even if those were just research results.
Response: I thank the reviewer for the powerful inquiry of inquisitive nature, Bravo. Our published data from my lab indicated that human alpha defensin 5 (DEFA5 alias HD5) is aberrantly expressed in the tissues of CC but not UC patients. We have further shown that DEFA5 is highly expressed in the colon crypt mucosa secreted by Paneth cell-like cells (PCLCs) or Crypt-cell like cells (CCLCs) in areas observed as ectopic “colon ileal metaplasia” that is consistent with the diagnosis of CC (PMID: 33690604 and PMID: 28817680, in this paper are reference nrs. 71 & 11). Preliminary characterization of this discovery revealed that detection of DEFA5 in IBD tissues, is a powerful tool to unambiguously differentiate CC from UC amongst IC patient cohort. The potential for DEFA5 as a diagnostic tool to discriminate UC from CC and IC into authentic CC or UC has been filed as US Patent No. 16/571,034, wipo.int, and the potential monoclonal antibodies to develop a reliable and clinically relevant sandwich ELISA is also patented as US Patent No. 16/622,259, https://patentimages.storage.googleapis.com/3d/f2/bd/5760a730bb31c5/WO2018237064A1.pdf. We have recently developed two mouse IgG monoclonal antibodies (mAbs) against DEFA5 and have tested at least two commercially available DEFA5 mAbs. The goal is to validate our antibodies compared to the available mAbs to develop a clinically relevant bioassay tool to definitively address the IBD diagnosis dilemma in clinical settings.
- Still some typos.
Response: I did run the English language Grammar and spelling engine and indicated no grammatic or spelling mistakes.
aem
04/06/2022
Reviewer 2 Report
In the revised version the author addressed some of the concerns raised by this reviewer, and the current version indeed is more focused. Nonetheless, couple suggestions for the author:
- Restructure the text. There are too many headings, and some of them could be combined into one section as subheadings. For instance, after Clinical diagnosis there are CD, UC, IC, etc. Could they be put under “clinical diagnosis and manifestation”or something similar. Also, “treatment and management” could cover several of the current headings.
- In the molecular diagnostic advances section, is there any other new progress? As with the development of multi-omics I am curious if there is any related research in IBD--even if those were just research results.
- Still some typos.
Author Response
Response to Reviewers
Medicina-1639493
Reviewer: Restructure the text. There are too many headings, and some of them could be combined into one section as subheadings. For instance, after Clinical diagnosis there are CD, UC, IC, etc. Could they be put under “clinical diagnosis and manifestation” or something similar. Also, “treatment and management” could cover several of the current headings.
Response: I sincerely thank the reviewer for outstanding job and for the rewarding comments and suggestions. As suggested, I have combined headings to single section with subheadings.
Reviewer: In the molecular diagnostic advances section, is there any other new progress? As with the development of multi-omics I am curious if there is any related research in IBD--even if those were just research results.
Response: Thank you for asking. Unfortunately, these studies are painstakingly slow, but we continue in the right path. We first analyzed well characterized unambiguous UC vs. CC colon tissue biopsies using Omics Data Analysis and microarray as training test and NanoString technology as an independent test set. Our first line discovery studies have been completed and published (see refs # 11,39,70,75 & 76) from my lab. Pre-clinical studies are underway and then, if successful, clinical trials will be initiated. Our ASSURED, the abbreviation - Affordable, Sensitive, Specific, User-friendly, Robust and Rapid, Equipment-free, and Deliverable approach, is an immunoreactivity that is specific, sensitive, linear, affordable, low risk, & less-invasive bioassay that can delineate subtypes of IBD during the patient first clinic endoscopy biopsy visit. We believe that indeterminate colitis will be wiped out completely.
Reviewer: Still some typos.
Response: I did run the English dictionary engine for typos and few spelling errors were found and corrected.

This manuscript is a resubmission of an earlier submission. The following is a list of the peer review reports and author responses from that submission.